# Study on g-C_3_N_4_/BiVO_4_ Binary Composite Photocatalytic Materials

**DOI:** 10.3390/mi14030639

**Published:** 2023-03-11

**Authors:** Pengfei Li, Yanqiu Hu, Di Lu, Jiang Wu, Yuguang Lv

**Affiliations:** 1College of Pharmacy, Jiamusi University, Jiamusi 154007, China; 2School of Stomatology, Jiamusi University, Jiamusi 154007, China

**Keywords:** BiVO_4_, g-C_3_N_4_, photocatalytic

## Abstract

Recent studies have shown that the composite of semiconductor photocatalytic materials and g-C_3_N_4_ can effectively inhibit photocatalytic carrier recombination and enhance the adsorption performance of the composite photocatalytic materials, so that the composite photocatalyst has stronger photocatalytic activity. In this paper, three kinds of graphitic carbon nitride photocatalyst g-C_3_N_4_ with different morphologies were prepared using the same precursor system by the chemical cracking method. After characterization and application, the sample with the most significant photocatalytic activity was selected and the g-C_3_N_4_/BiVO_4_ heterostructure was synthesized by the simple solvent evaporation method, then the photocatalytic experiment was carried out. The results show that, when the content of BiVO_4_ in the composite sample is 1%, the photocatalytic activity of RhB was the highest, and the degradation rate could reach 90.4%. The kinetic results showed that the degradation of RhB was consistent with the quasi-primary degradation kinetic model. The results of the photocatalytic cycle experiment show that the photocatalytic performance remains unchanged and stable after four photocatalytic cycles. The existence of a g-C_3_N_4_/BiVO_4_ binary heterojunction was confirmed by UV/Visible diffuse reflection (UV-DRS) and photoluminescence (PL) experiments. Owing to the Z-type charge process between BiVO_4_ and g-C_3_N_4_, efficient carrier separation was achieved, thus enhancing the photocatalytic capacity. This work provides a new idea for the study of heterojunction photocatalytic materials based on g-C_3_N_4_.

## 1. Introduction

Semiconductor photocatalysts have become increasingly well known in recent decades. They have been paid attention because of their advantages, such as being green, no pollution, high efficiency, energy saving, stability, and low price. Photocatalysis technology has shown excellent performance in improving the environment, making it one of the research hotspots in pollutant degradation [1,2,3,4]. Currently, the most used catalysts are TiO_2_ [5,6,7] and ZnO [8,9,10,11], among others. However, their catalytic performance is not efficient. In visible light, the utilization rate of light is low, which limits its application in real life scenarios. Therefore, a new type of photocatalyst needs to be prepared to compensate for the shortcomings of conventional catalysts.

g-C_3_N_4_ [12,13,14] has the advantages of narrow band gap, low cost, acid and alkali resistance, and environmental protection. Therefore, g-C_3_N_4_ has become a hot research topic in recent years, although monomeric g-C_3_N_4_ has some defects, such as low utilization of sunlight and fast photogenerated carrier complexation. These defects hinder its application in the process of environmental treatment. In order to improve the defects existing in g-C_3_N_4_, as well as to improve the photocatalytic activity of g-C_3_N_4_, modification research has been carried out. Different methods such as doping [15,16,17], morphology control [18,19], and surface loading [20] can be used.

BiVO_4_ [21,22,23] is a potential functional semiconductor material discovered in recent years. Thanks to its advantages such as a wide source of constituent elements; excellent physical, chemical, and thermal stability; being non-toxic and harmless; and narrow band gap, BiVO_4_ can be excited to generate photogenerated electrons and holes under visible light, so as to realize more effective utilization of solar energy. It has recently become one of the hot spots in the field of photocatalysis. However, the electrons in BiVO_4_ move at a slow rate, resulting in about 60–80% of the resulting charge carriers recombining before they reach the surface. The photogenerated electron hole pairs in BiVO_4_ materials are difficult to separate and easy to recombine. At present, no single BiVO_4_ material can achieve its theoretical photoelectric conversion efficiency. However, owing to the particularity of the band structure, the heterojunction composite semiconductor photocatalyst can greatly improve the separation efficiency of electron–hole pairs and produce responses to most spectral frequencies, so as to improve the photocatalytic efficiency.

To sum up, three kinds of g-C_3_N_4_ with different morphologies were prepared using urea as the precursor system. The best morphology was g-C_3_N_4_ by B characterization and performance analysis, the Z-type heterojunction GCB composite samples were successfully prepared by combining with BiVO_4_, and its catalytic activity was determined by the photocatalytic degradation of rhodamine B (RhB) under visible light irradiation. The photocatalytic reaction mechanism is also presented. This study provides a reference for the further development and reference of g-C_3_N_4_.

## 2. Materials and Methods

### 2.1. The Preparation of g-C_3_N_4_

Here, 10 g of melamine was placed into an alumina crucible to form a semi-closed space. The crucible was placed in a muffle furnace under the protection of an air atmosphere; after heating to 500 °C for 2.5 h, deamination polymerization was carried out. After cooling to room temperature, yellow bulk g-C_3_N_4_ samples were obtained, named g-C_3_N_4_-b.

### 2.2. The Preparation of Sheet g-C_3_N_4_

The g-C_3_N_4_-b was ground into powder, 10 mL of 2 mol/L H_3_PO_4_ was added, and it was stirred at room temperature for 30 min, followed by ultrasonic stirring for 30 min. The mixture was transferred to a reaction kettle, heated at 160 °C in a constant temperature blast drying oven for 4 h, then cooled to room temperature, cleaned three times with deionized water and anhydrous ethanol, and dried for 8 h to obtain the sheet g-C_3_N_4_ sample—named g-C_3_N_4_-s.

### 2.3. The Preparation of g-C_3_N_4_ Tube

The g-C_3_N_4_-s was ground into powder and 10 mL of 2 mol/L NH_3_·H_2_O was added. It was dispersed uniformly by ultrasound for 30 min, then transferred to a reaction kettle, stored at 180 °C in a constant temperature blast oven for 4 h, cooled to room temperature, and washed three times with deionized water and anhydrous ethanol to obtain the tubular g-C_3_N_4_ sample—named g-C_3_N_4_-t.

### 2.4. Three Kinds of g-C_3_N_4_ Morphologies Were Evaluated in Photocatalytic Experiments

The photocatalytic degradation of photocatalysts was evaluated using rhodamine B as an organic dye model. Here, 0.5 g g-C_3_N_4_-b, g-C_3_N_4_-s, and g-C_3_N_4_-t were weighed into three beakers and 100 mL of 20 mg/L RhB solution was added to each beaker; RhB solution was used as the source of pollution and the samples were magnetically stirred in the dark for 30 min. After adsorption equilibrium, a xenon lamp of 350 watts was irradiated as the visible light source. The absorbance of 5 mL samples was measured every 30 min. The maximum absorption wavelength was 553 nm and it was measured with an ultraviolet visible spectrophotometer.
(1)$$\mathrm{Degradation}\;\mathrm{rate}\;\mathrm{of}\;\mathrm{RhB}\;=\;\frac{C_{t}}{C_{0}}\;\times \;100\%$$
where C_t_ is the concentration of RhB measured after a period of reaction and C_0_ is the concentration at the beginning of the reaction. The degradation rate of Rhodamine for the three samples was calculated, and the sample with the highest degradation rate was selected for reserve.

### 2.5. The Preparation of BiVO_4_

Here, 0.25 g Bi (NO_3_)_3_·5H_2_O was dissolved in 10 mL of 2 mol/L H_3_PO_4_ solution and 0.12 g of NH_4_VO_3_ was dissolved in 10 mL of 2 mol/L NH_3_·H_2_O solution. The two were slowly mixed so that they were stirred evenly, and the pH of the mixed solution was adjusted to 7. After the above process, a uniform clear yellow suspension was formed. The solution was transferred to the reactor and kept in a constant temperature blast oven at 160 °C for 8 h. It was then cooled to room temperature, subjected to high-speed centrifugation, washed several times with deionized water and anhydrous ethanol, and dried at 60 °C for 8 h, thus BiVO_4_ was obtained.

### 2.6. The Preparation of Z-Type Heterojunction g-C_3_N_4_/BiVO_4_

The g-C_3_N_4_ sample with the best 5 g degradation effect was dissolved in 10 mL of ethanol solution, and the g-C_3_N_4_ was evenly dispersed by ultrasonic treatment. BiVO_4_ with mass fractions of 0.3%, 0.5%, 1%, 3%, and 5% was added. After continuous agitation, the pH of the solution was adjusted to 7. Cooling to room temperature, high-speed centrifugal treatment, centrifugation with deionized water and anhydrous ethanol, washing for several times, and drying at 60 °C for 8 h were carried out to obtain Z-type heterojunction g-C_3_N_4_/BiVO_4_, named GCB-x (x = 0.3%, 0.5%, 1%, 3%, and 5%, respectively). When testing its photocatalytic performance, the experimental steps are the same as in Section 2.4.

### 2.7. Characterization Method

The information of substance composition and crystal structure can be obtained by XRD characterization; the model of XRD equipment used in this study is XDR-6100 and the scanning range is 10°–80°. The infrared spectrometer is IS10 with a scanning range of 500–4000 cm^−1^. SEM uses secondary electronic signal imaging to observe the surface morphology and structure of the sample. This article is used for research. XPS can determine the composition and state of atoms or ions in the surface layer of an unknown sample. The XPS used in this study are AXIS ULTRA devices from Shimadzu, Japan. The instrument used was a Nicolet 5700 infrared spectrometer to test the molecular structure inside the material. In the test process, KBr and the tested sample were mixed and ground at a ratio of 100:1, and then pressed to test the transmittance. Photoluminescence (PL) is the process by which a substance absorbs a photon and radiates it back. PL spectrum can characterize the defects, impurities, and luminescence properties of semiconductor materials. Fluoro Max-4p is the PL spectrometer used in this study, with a Xenon lamp with an excitation source of 150 W and excitation wavelength of 425 nm. A UV-4100 UV/Visible spectrophotometer (Hitachi, Tokyo, Japan) was used to obtain the visible diffuse spectra (UV/Vis diffuse reflectance spectra, DRS) to test the absorbance change in the powder sample with wavelength change; the scanning range is 200 nm–800 nm.

### 2.8. Evaluation of the Photocatalytic Mechanism

The light absorption characteristics of the samples were studied by solid UV/Vis DRS. This method was used to analyze the displacement of the side band and the change in the forbidden band gap between them. The band gap of the sample can be calculated using the formula below.
αhυ = A (hυ − Eg)^n/2^(2)
where α is the absorption coefficient, hυ is the photon energy, A is a constant equal to 1, and Eg is the band gap energy. The values of n are 4 for indirect transition and 1 for direct transition [24]. The valence and conduction band potentials of g-C_3_N_4_ and BiVO_4_ can be calculated using their electronegativity with the following empirical equations.
E_VB_ = X + 0.5Eg − Ee(3)
E_CB_ = E_VB_ − Eg(4)

Here, E_CB_ and E_VB_ represent the conduction band edge and valence band edge, respectively; X is the absolute electronegativity of the semiconductor; and Ee is a measurement scale factor of the redox level of the reference electrode relative to the absolute vacuum scale [25].

## 3. Results

Figure 1a shows massive g-C_3_N_4_-b, with many layers and serious agglomeration and a small specific surface area, which is not conducive to the photocatalytic reaction. Figure 1b shows g-C_3_N_4_-s, which has thin lamella and a relatively uniform distribution. Figure 1c shows g-C_3_N_4_-t. It can be seen that the rough surface and large specific surface area provide more active sites, which is conducive to the construction of the semiconductor heterojunction and improving the catalytic activity of the catalyst [26].

The XRD pattern of the g-C_3_N_4_ photocatalyst is shown in Figure 2a. It shows that g-C_3_N_4_-t, g-C_3_N_4_-s, and g-C_3_N_4_-b all have two characteristic peaks. The diffraction peak of g-C_3_N_4_ at 27.5° is the typical superposition reflection between layers of the conjugated aromatic hydrocarbon system, and the small peak at about 13.1° corresponds to the (100) diffraction plane of g-C_3_N_4_. It can also be seen from Figure 2a that the diffraction intensity of plane (100) and (002) in g-C_3_N_4_-s is weaker than that of plane g-C_3_N_4_-b and g-C_3_N_4_-t, which may be because of the fact that the CO_2_ produced by C and O in urea during the long calcination of t inhibits the growth of the crystal surface, thus forming structural defects in the sample. Figure 2b shows the FT-IR spectrum of the prepared sample. The spectra of pure g-C_3_N_4_-t, g-C_3_N_4_-s, and g-C_3_N_4_-b all showed the characteristic peak of g-C_3_N_4_. The results showed that multiple peaks in the range of 1230–1650 cm^−1^ were caused by C=N: 1637 cm^−1^ was attributed to C=N and 1235 cm^−1^–1574 cm^−1^ was attributed to aromatic C-N. The wide peak centered on 3180–3440 cm^−1^ was attributed to N-H. At 1574 cm^−1^, the absorption peak strength of g-C_3_N_4_-t is higher than that of g-C_3_N_4_-s and g-C_3_N_4_-b, indicating that g-C_3_N_4_-t has good crystallinity and structural integrity, which is consistent with the XRD results. Figure 2c shows that the three samples have strong fluorescence emission peaks at 547 nm and g-C_3_N_4_-b has a higher fluorescence intensity, while the g-C_3_N_4_-t form has a lower fluorescence intensity. The results showed that the position of the emission peak remained unchanged after morphology regulation, but the fluorescence intensity was relatively reduced. Therefore, g-C_3_N_4_-t fluorescence intensity was the lowest, facilitating the separation of photogenerated carriers and effectively improving the photocatalytic activity.

Figure 3a shows the degradation effect of pollutants. With the passage of light time, the characteristic absorption peak of RhB in the UV/Visible absorption spectrum weakens synchronously, and no new absorption peak appears in the whole spectrum. When the characteristic absorption peak decreases, it indicates that the concentration of the RhB specific hair group decreases, and the solution gradually becomes transparent, so it can be proved that the pollutant is decomposed at a certain rate. Figure 3b shows the comparison curve of the degradation effect of the synthetic sample on RhB, and it can be seen that the RhB aqueous solution does not easily degrade naturally under visible light radiation, while g-C_3_N_4_-t showed the best photocatalytic activity. For the data results in Figure 3b, the relationship curve of −ln (C_t_/C_0_)-t is used to represent the first-order reaction kinetics model of the degradation effect in Figure 3c. As shown in Figure 3c, the fitted relationship curve of −ln (C_t_/C_0_)-t is a straight line, and the correlation coefficients R^2^ of the line system are all greater than 0.98, which proves that the photodegradation reaction of this series of synthesized samples follows the first-order reaction kinetics law. As the apparent kinetic constant of the quasi-first-order photocatalytic degradation reaction is positively correlated with its photocatalytic activity, under the same experimental conditions, the photocatalytic activity order of the sample is g-C_3_N_4_-t > g-C_3_N_4_-s > g-C_3_N_4_-b.

The morphology, crystal structure, functional group, and fluorescence intensity of three different kinds of g-C_3_N_4_ were tested and analyzed in detail. It can be seen from the characterization results that the modified functional groups are not damaged and the crystal structure is not significantly changed, but g-C_3_N_4_-t has a large specific surface area and can provide more active sites. The fluorescence intensity of g-C_3_N_4_-t was the lowest, and the photocatalytic activity was improved effectively. The photocatalytic results showed that g-C_3_N_4_-t had the best degradation effect. Therefore, g-C_3_N_4_-t was selected for follow-up study.

Figure 4 is a composite sample of g-C_3_N_4_ and BiVO_4_. It is hereinafter referred to as GCB. The rough surface of g-C_3_N_4_-t can provide more active sites and promote semiconductor recombination to form a heterojunction, which is conducive to better absorption of visible light and separation of photogenerated charge, thus improving the photocatalytic efficiency.

For the g-C_3_N_4_ sample, it can be clearly seen that the diffraction peak intensity is the highest at 27.4°, corresponding to the (002) crystal plane, highly consistent with standard cards (JCPDS 87-1526). BiVO_4_ has several very obvious characteristic peaks, at 24.52°, 29.08°, 34.49°, and 39.78°. The peak position is basically consistent with the monoclinic crystal of BiVO_4_ standard card (JPCDS14-0685). With the increase in BiVO_4_ [27] content, the intensity of the diffraction peak gradually increases, indicating that the two are successfully recombined. Figure 5b shows that g-C_3_N_4_ at 810 cm^−1^ is attributed to the bending vibration of the hepazine ring system, 1637 cm^−1^ is attributed to V_C=N_, and 1235 cm^−1^–1574 cm^−1^ is attributed to aromatic V_C-N_. The wide peak centered on 3180–3440 cm^−1^ was attributed to V_N-H_. The corresponding peaks of BiVO_4_ at 742 cm^−1^ and 842 cm^−1^ are attributed to V_σas3_ (VO_4_) and V_σa1_ (VO_4_). The peak intensities of V_σas3_ (VO_4_) and V_σa1_ (VO_4_) in the GCB composite sample increased slightly with the increase in BiVO_4_ content, indicating the existence of two phases in the composite sample. As can be seen from Figure 5c, the emission peaks of BiVO_4_ and GCB composite samples appear at 541 nm. The fluorescence intensity of BiVO_4_ and g-C_3_N_4_ is higher than that of the GCB composite sample, indicating that the photogenerated charges of GCB are more easily separated. It can be seen from the figure that the fluorescence intensity changes with the increase in BiVO_4_ content. The sample fluorescence intensity is lowest when the doping amount is GCB-1%. It is speculated that the excess BiVO_4_ may cover the active site and reduce the reactive groups.

In order to further characterize the surface chemical composition of g-C_3_N_4_, BiVO_4_, and g-C_3_N_4_/BiVO_4_ composite catalysts and the interaction between g-C_3_N_4_ and BiVO_4_ in composite heterojunction materials, we performed XPS tests on pure g-C_3_N_4_, BiVO_4_, and GCB-1%. As can be seen from the full spectrum of XPS in Figure 3a, the characteristic absorption peaks on the surface of g-C_3_N_4_ are C 1s and N 1s, and those on the surface of BiVO_4_ are Bi 4f, V 2p, and O 1s. These absorption peaks are present on the surface of the GCB-1% composite. In g-C_3_N_4_/BiVO_4_ composites, the binding energy of C 1s is higher than that of g-C_3_N_4_. As shown in Figure 6c, the characteristic peaks of g-C_3_N_4_/BiVO_4_ in the composite materials all shifted, indicating that g-C_3_N_4_ interacts strongly with BiVO_4_ during the formation of the composite catalyst, which increases the binding energy of C 1s and N 1s. As shown in Figure 6d, Bi 4f has two characteristic peaks, and the binding energies of Bi 4F_7/2_ and Bi 4F_5/2_ are 159.2 eV and 164.5 eV, respectively. As shown in Figure 6e, V 2p has two characteristic peaks, and the binding energies of V 2P_1/2_ and V 2P_3/2_ correspond to 516.8 eV and 523.7 eV, respectively. As shown in Figure 6f, O 1s has three characteristic peaks. Moreover, 529.5 eV represents the lattice O in BiVO_4_, and 530.2 eV and 532.3 eV correspond to hydroxyl oxygen and O adsorbed in water on the catalyst surface, respectively. Through the analysis of element species, it was further proved that the GCB composite sample was successfully prepared.

Figure 7a shows that the GCB composite samples display a red shift phenomenon, and the light absorption capacity is obviously improved. With the increase in BiVO_4_ content, the red shift was more obvious. Eg corresponding to g-C_3_N_4_ and GCB composite samples was calculated according to the formula. It can be seen from the figure that the absorption band of g-C_3_N_4_ is about 470 nm, Eg = 2.63 eV. The absorption band of the GCB composite sample is about 510 nm, Eg = 2.43 eV. The results show that the preparation of GCB composite samples reduces the band gap width and improves the visible light response ability of the composite samples.

GCB (m) is a simple physical mixture of g-C_3_N_4_ and BiVO_4_, and GCB is a composite catalyst with a heterojunction prepared by the hydrothermal method, Figure 8a shows that, after 30 min reaction in the dark, GCB has a strong adsorption capacity. This is because, in the process of doping, ultrasound can evenly disperse g-C_3_N_4_-t in deionized water, so that GCB has a large specific surface area. The degradation efficiency of GCB (m) was significantly lower than that of GCB during the photoreaction, which was caused by the formation of the Z-type heterojunction between the GCB composite samples. Figure 8b shows that the photocatalytic activity of g-C_3_N_4_-t is low with the increase in time after the dark reaction for 30 min and then light reaction for 300 min. D_RhB_ = 33.4%. Even though g-C_3_N_4_-t can absorb visible light, the photoelectron–hole pair recombination rate is high, which inhibits their photocatalytic activity. When different BiVO_4_ contents were added, the D_RhB_ of the GCB-0.3% sample was 59.6%, that of the GCB-0.5% sample was 63.3%, and that of the GCB-1% sample was 90.4%. However, when the content of BiVO_4_ was increased further, the degradation rate did not increase, but rather decreased. It may be that excessive doping of BiVO_4_ may attach to the surface of g-C_3_N_4_, thus reducing the active site and specific surface area of the surface, resulting in lower degradation efficiency of RhB. Therefore, the degradation rate of RhB reaches 90.4% when the doping amount is GCB-1% under illumination for 300 min, and it has the best catalytic activity.

Table 1 shows that GCB-1% has the highest k value, k = 0.00434 min^−1^, indicating that the sample has the best degradation effect on RhB. The correlation coefficients R^2^ of the GCB composite samples are all greater than 0.9, and the results show that the samples conform to the first-order reaction kinetic equation.

Figure 9a shows that, after four cycles, the D_RhB_ of the GCB-1% sample decreased somewhat, but the D_RhB_ still remained above 80%, proving that the structure of the prepared GCB sample was not damaged and still had good activity, indicating that the GCB composite sample has good repeatability. As shown in Figure 9b, D_RhB_ = 90.4% of the GCB-1% sample, while the photocatalytic activity did not change significantly when IPA was added. After EDTA and VC were added, the photocatalytic activity was reduced, with D_RhB_ = 32.6% and D_RhB_ = 51%. Therefore, the experiment shows that h^+^ and ·O^2−^ are the main active species in the catalytic system of GCB, among which ·O^2−^ is more important to the degradation of RhB.

Figure 10 is a diagram of the photocatalytic reaction mechanism.The crystal structure of BiVO_4_ is composed of octahedral layers arranged alternately, which is conducive to enhancing the separation of e^−^– h^+^. Table 2 shows that the gap between CB_g-C3N4_ and CB_BiVO4_ (1.62 eV) is much larger than that between VB_BiVO4_ and VB_g-C3N4_ (1.37 eV), so the e^−^ of CB_BiVO4_ is easily transferred to VB_g-C3N4_. As a result, the e^−^ of CB_BiVO4_ and h^+^ of VB_g-C3N4_ are easily recombined, and more e^−^ reduced oxygen molecules on CB_g-C3N4_ produce ·O^2−^. Therefore, the photocatalytic activity of GCB samples was significantly enhanced. It can be concluded that g-C_3_N_4_ /BiVO_4_ is a Z-type heterojunction photocatalyst.

## 4. Conclusions

Three kinds of g-C_3_N_4_ with different morphologies were prepared using urea as the precursor system. The best morphology was g-C_3_N_4_-t, by B characterization and performance analysis, and the Z-type heterojunction GCB composite samples were successfully prepared by combining with BiVO_4_, and were characterized by various methods. In order to test the photocatalytic activity of GCB-1%, the degradation experiment of RhB aqueous solution was carried out. The stability and photocatalytic mechanism of Z-type heterojunction GCB-1% composite samples were discussed. TEM shows that doping BiVO_4_ increases the contact area, improves the separation efficiency of photogenerated charge, and increases the photocatalytic degradation efficiency. XRD shows that the prepared BiVO_4_ is monoclinic crystal phase and the prepared g-C_3_N_4_ is tetragonal phase, indicating that the two are successfully compounded. FT-IR shows that the peak intensities of V_σas3_ (VO_4_) and V_σa1_ (VO_4_) increase slightly with the increase in BiVO_4_ content, indicating the existence of g-C_3_N_4_ and BiVO_4_ phases in the composite sample. PL shows that, when the doping amount is GCB-1%, the fluorescence intensity is the lowest and the photocatalytic ability is the strongest. PL shows that, with the doping amount of GCB-1%, the fluorescence intensity is the lowest, that is, the electron–hole recombination rate is the highest. XPS indicated that the diffraction peaks of C 1s and N 1s in GCB-1% moved in the direction of higher binding energy, and the diffraction peaks of O 1s and Bi 4f moved in the direction of lower binding energy. These changes indicated that g-C_3_N_4_ interacted with BiVO_4_ during the formation of g-C_3_N_4_/BiVO_4_ composites. According to the UV/Vis results, the visible light absorption intensity of GCB composite samples increased, while the gap between the GCB composite samples decreased, indicating that the doped composite samples were more favorable to the improvement in photocatalytic activity. When the doping amount is GCB-1%, D_RhB_ = 90.4% is the highest. The cyclic stability test shows that the GCB-1% composite sample has good repeatability. The results of free radical capture experiments showed that h^+^ and ·O^2−^ were the main active components in the GCB catalytic system, and ·O^2−^ played a decisive role in the degradation of RhB.

## Figures and Tables

**Figure 1 micromachines-14-00639-f001:**
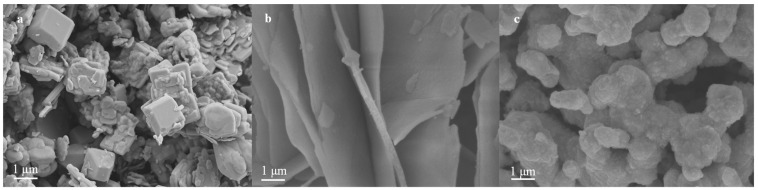
SEM diagram of (**a**) g-C_3_N_4_-b; (**b**) g-C_3_N_4_-s; and (**c**) g-C_3_N_4_ -t.

**Figure 2 micromachines-14-00639-f002:**
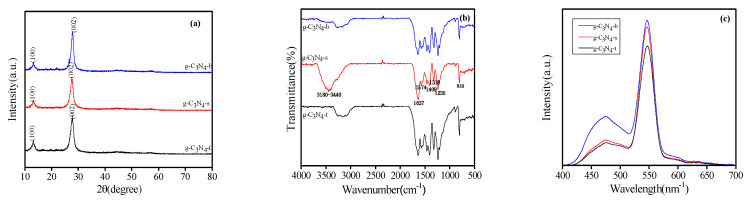
(**a**): XRD diagram of g-C_3_N_4_-b, g-C_3_N_4_-s, and g-C_3_N_4_-t. (**b**): FT-IR diagram of g-C_3_N_4_-b, g-C_3_N_4_-s, and g-C_3_N_4_-t. (**c**): PL diagram of g-C_3_N_4_-b, g-C_3_N_4_-s, and g-C_3_N_4_-t.

**Figure 3 micromachines-14-00639-f003:**
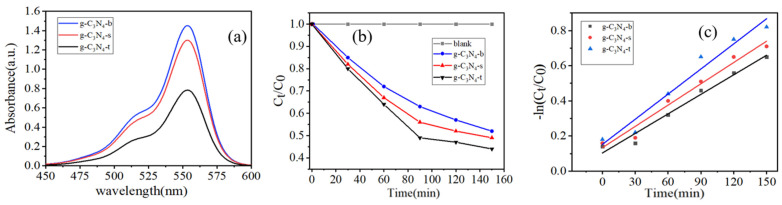
(**a**): Change in absorbance of Rhodamine B degraded by photocatalyst with time, (**b**): the comparison curve of the degradation effects of the synthesized sample on RhB, (**c**): the first-order reaction kinetics curve of the degradation effects of the synthesized sample on RhB.

**Figure 4 micromachines-14-00639-f004:**
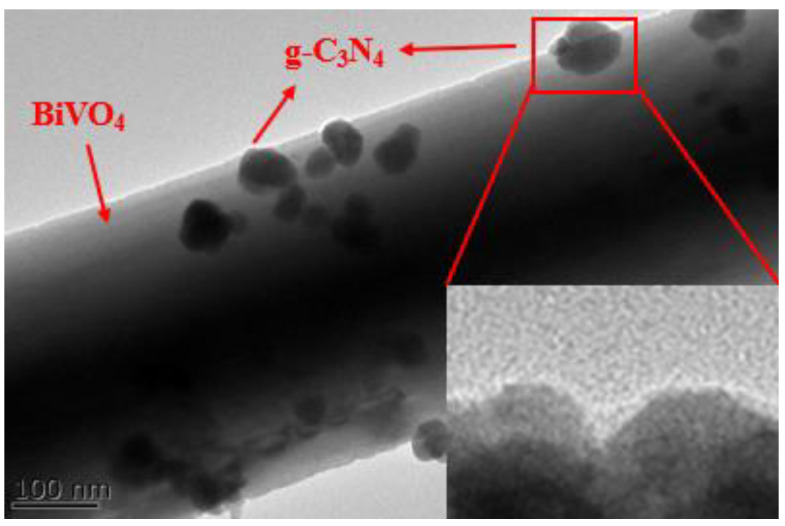
TEM diagram of the composite sample of g-C_3_N_4_ and BiVO_4_.

**Figure 5 micromachines-14-00639-f005:**
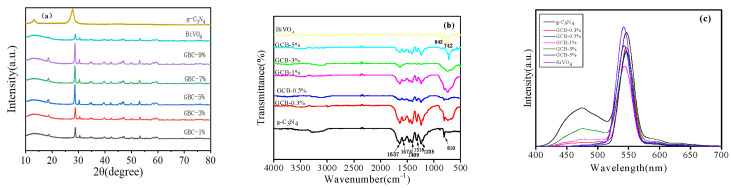
XRD diagram of (**a**) g-C_3_N_4_, BiVO_4_, GCB composite sample. FT-IR diagram of (**b**) g-C_3_N_4_, BiVO_4_, GCB composite sample. PL diagram of (**c**) g-C_3_N_4_, BiVO_4_, GCB composite sample.

**Figure 6 micromachines-14-00639-f006:**
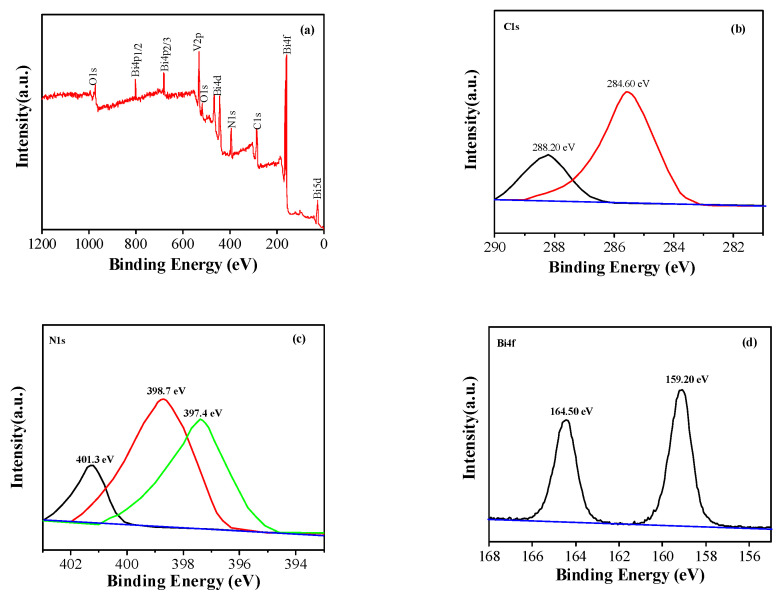
XPS patterns of BiVO4/g-C3N4 samples: full spectrum (**a**), C 1s (**b**), N 1s (**c**), Bi 4f (**d**), V 2p (**e**), and O 1s (**f**).

**Figure 7 micromachines-14-00639-f007:**
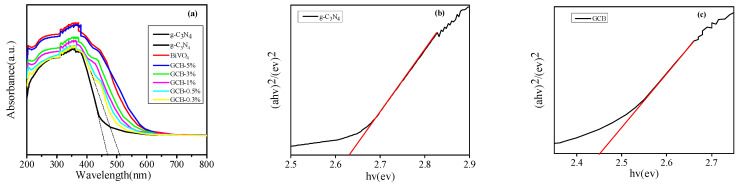
UV/Vis (**a**) and band gap of (**b**) g-C_3_N_4_ and (**c**) GCB composite sample.

**Figure 8 micromachines-14-00639-f008:**
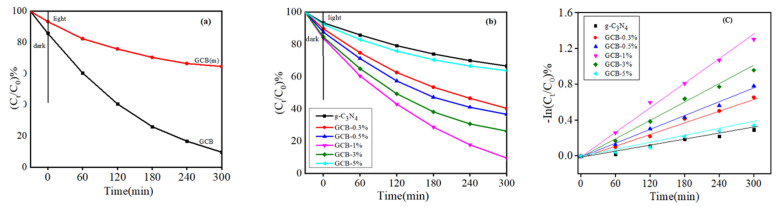
(**a**) Photocatalytic degradation curves of GCB (m) and GCB; (**b**) photocatalytic degradation curves of different doping amounts of BiVO_4_; (**c**) first-order reaction constants of GCB photocatalytic degradation of RhB.

**Figure 9 micromachines-14-00639-f009:**
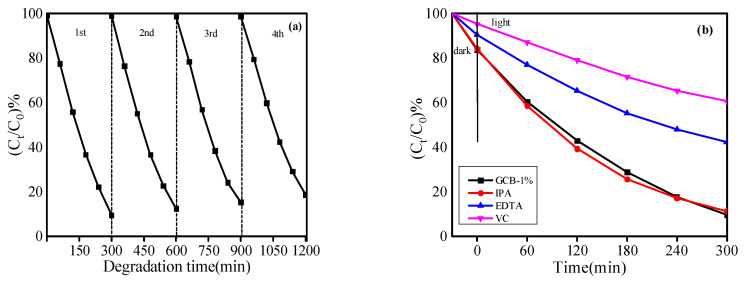
(**a**) Stability of GCB-1% photocatalytic degradation of RhB dye; (**b**) the results of the reactive-group-capture experiment for the photocatalytic degradation of RhB by the GCB-1% sample.

**Figure 10 micromachines-14-00639-f010:**
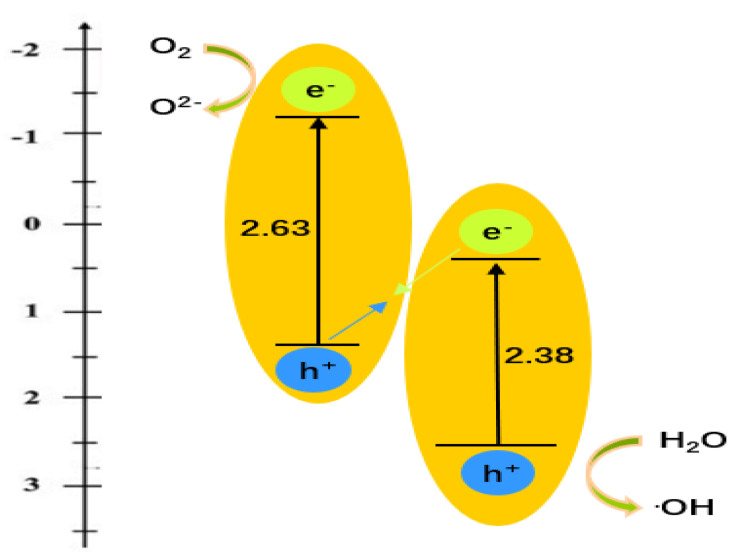
Diagram of the photocatalytic mechanism.

**Table 1 micromachines-14-00639-t001:** Linear fitting data of photocatalytic reaction kinetics of GCB composite samples.

The Sample Name	The Regression Equation	k	R^2^
g-C_3_N_4_	Y = 0.0010 − 0.014 x	0.00099	0.9784
GCB-0.3%	Y = 0.0022 − 0.0179 x	0.00219	0.9915
GCB-0.5%	Y = 0.0025 − 0.0134 x	0.00251	0.9932
GCB-1%	Y = 0.0044 + 0.0207 x	0.00434	0.9959
GCB-3%	Y = 0.0033 − 0.0019 x	0.00328	0.9941
GCB-5%	Y = 0.0012 − 0.0174 x	0.00117	0.9808

**Table 2 micromachines-14-00639-t002:** Band gap energy (eV) and conduction band and valence band potential (V) of g-C_3_N_4_ and BiVO_4_.

Semiconductors	Eg (eV)	CB (eV)	VB (eV)
g-C_3_N_4_	2.63	−1.15	1.48
BiVO_4_	2.38	0.47	2.85

## Data Availability

Subsequent articles need to be used and will not be made public for the time being.

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
