# Peer review of "Study on g-C3N4/BiVO4 Binary Composite Photocatalytic Materials"

_micromachines, 2023, doi:10.3390/mi14030639_

Round 1
Reviewer 1 Report (Previous Reviewer 1)
The authors have done a good job and the article can be accepted.
Author Response
非常感谢您的回复
Reviewer 2 Report (Previous Reviewer 2)
The manuscript (Manuscript ID. micromachines-2274455) reports the preparation, structural analysis and the photocatalytic studies of g-C3N4 prepared by 3 different ways. They also investigated their most efficient g-C3N4 composite modified by BiVO4.
The authors have corrected many errors. The text is mostly technically acceptable, but still contains many grammatical errors, making it difficult to understand (especially sections 2.2-2.4). I still maintain that it should be read by a native English-speaking reviewer.
Some previously reported errors were not or only partially corrected. These are:
A space must be used before units of measurement.
The section 2.2-2.4. should be reworded.
Figure 8. and Figure 9.: The „%” is missing from Y-axis captions.
The caption of Table 2 is still incorrect. „energy band, conduction band and valence band of g-C3N4 and BiVO4” should be replaced by, for example „Band gap energy (eV) and conduction band and valence band potential (V) of g-C3N4 and BiVO4”
In several cases, the figure and table captions are professionally inaccurate, incomplete and also require grammatical correction.
Line 320: „h+” (hole) should be written instead of „H+” (hydrogen ion).
Several sub and superscripts have been corrected but some remained incorrect in lines 373, 402, 410, 428 and 432.
And there are some mistakes in the new text:
Line 11: g-C3N4 is not pure carbon so it cannot be a „carbon allotrope”.
Line 93-94: Xe-lamps also emit in the UV range, not only in the visible range. Did you use a filter?
Line 226: „(a)” is not necessary because there is no (b).
The Tauc method is not suitable for calculating the energy gap for composites only for the single semiconductors.
Line 150 and 426-427: Ref. [24] is not a relevant paper for Tauc method. The original Tauc work should be referred: https://doi.org/10.1002/pssb.19660150224
There are two references [25]. In the case of the 2nd one, the parameters are incorrect (doi:10.1088/1755-1315/898/1/012025). However, there are no reference 27 despite the reference to this in line 169.
Why is the intercept not zero in Figures 3c and 8c and in Table 1? The question arises as to what the authors considered to be C0. The authors correctly defined C0 in lines 100-101 as "C0 is the concentration at the beginning of the reaction". The question is whether the authors calculated with concentrations before or after adsorption. The intercepts in Table 1 and Figure 8c are different.
The rate constants are usually denoted by lower case k not upper case K (Table 1).
Author Response
Please see the attachmen

This manuscript is a resubmission of an earlier submission. The following is a list of the peer review reports and author responses from that submission.
Round 1
Reviewer 1 Report
This manuscript describes synthesis of composite g-C3N4/BiVO4 for the photocatalytic degradation. Such composites have long been known, in general, the work looks absolutely ordinary. In addition, the introduction does not clearly state the purpose and objectives of the work. The text is written very carelessly, with a lot of typos, the figures are very small and of poor quality.
The most important note: the authors in the text contradict themselves. In line 180, they write that bulk g-C3N4-b was selected for Z-scheme creation with BiVO4 and justify this by its highest activity in dye degradation (Fig. 3). However, then, in lines 183-190, the authors write that a tubular g-C3N4-t sample was chosen to create composites, and they write about the same in the experimental part (paragraph 2.6).
I think the authors should deal with this remark and then resubmit the work. In its present form, the article cannot be accepted for publication.
Author Response
“请看附件。”

Reviewer 2 Report
The authors prepared a g-C3N4/BiVO4 composite and applied it to the heterogeneous photocatalytic degradation of RhB under visible light illumination.
In the last decade, several papers have been published in which g-C3N4/BiVO4 composite was prepared and applied to the heterogeneous photocatalytic degradation of RhB under visible light illumination, similar to the present manuscript (https://doi.org/10.1039/C6RA27766G, https://doi.org/10.1016/j.jtice.2018.10.011, https://doi.org/10.1021/acs.langmuir.7b00893, https://doi.org/10.1063/1.5090410, https://doi.org/10.1016/j.molcata.2016.08.025, https://doi.org/10.1016/j.apcatb.2018.04.056, https://doi.org/10.1007/s11595-015-1128-3, https://doi.org/10.1016/j.jcis.2018.08.071). The authors do not mention any of these in the introduction, although this would be very important. To accept the manuscript, it should also be made clear how it differs from the others, and what is novelty in it.
Further comments:
Lines 11-15: The 3rd sentence of abstract should be split.
Lines 19-20: … „which can effectively improve the separation efficiency of photogenerated electronhole pairs and inhibit their complexation.” „improve the separation efficiency” and „inhibit their complexation” are equivalent in this case. One of them should be deleted. „complexation” is not th ebest word here. I’d prefer „recombination”.
The last sentence of the abstract should be split and rephrased.
Line 35: The term „normal light” should be replaced by „visible light” or „sunlight”.
Line 41: „full and” should be deleted.
Line 43: „however” with low case
Line 68: Delete unnecessary 'T'.
A space must be used before units of measurement.
Line 70: The word "process" should be replaced by another word.
Line 78: „A slice …” should be corrected.
The section 2.4. is not clear. It should be reworded.
Line 96: A suspension cannot be clear.
Line 118: „before and after” is correctly “after and before”.
In the "Materials and Methods" section, a sequence of instructions is used instead of complete sentences.
Lines 125-128, lines 162-172, lines 193-195, …: „Figure 1a” instead of „Figure (a)”…
The abbreviations of composites („g-C3N4-b”, „g-C3N4-s”, „g-C3N4-t”) should be defined in section 2.2-2.4.
In figures 2, 3 and 5: It would be better to assign the same colours to the same composites.
Line 157: „The original degradation maps …” – This caption is not precise enough.
Figure 3a: Why is the baseline of absorption spectra of the system containing g-C3N4-s and g-C3N4-t catalysts so high? Perhaps these have been shifted?
Figures 3b and 3c: After 150 min the measured c/c0 values are between about 0.42 and 0.54, buti n the Figure 3c their negative logaritms are between 0.25 and 1.8 instead of 0.61 and 0.87. How can it be possible? Which is the correct order of the activity? The order in Figure 3b is b > t > s, while in Figure c it is b > s > t.
Line 166: „transparent” or „colorless”?
Line 170: Space is missing: „… activity. For…”.
Line 171 and 173: „−ln(Ct/C0)-” instead of „-ln (Ct/C0)-” and „R2” instead of „R”. Ct should be defined.
Line 185: „different kinds of” instead of „KINDS”.
Lines 180-181 contradict lines 190-191.
Figure 4a ang Figure 4b shows the similar sample with different magnification. The figure and the caption and text (lines 193-199) are not consistent.
Several subscripts and superscripts have been omitted or the upper and lower case letters have been reversed in the following lines: 180, 231-236, 246, 262, 268, 288, 292, 295, 303-308, 311, 350, 375, 401. Some of the most serious mistakes: „E -„ as a symbol of electron, „H +” as a symbol of hole (h+).
Figure 8. and Figure 9.: All Y-axis captions are wrong. „(Ct/C0)%” or „−ln(Ct/C0)” should be used instead of „(C0/Ct)%”. The numbers are missing near the Y-axis in Figure 8c.
"GCB (m)" and "Z-GCB" have not been defined. It is therefore difficult to interpret the text after Figure 8.
In the dark, adsorption takes place, not degradation. Therefore, lines 258-261 should be modified accordingly.
How do you make sure that the adsorption equilibrium is reached after 30 minutes?
Lines 268-270: „GCB” instead of „CBC”
The caption of Table 2 is incorrect. Where do the bandgap energy, CB and VB potential values come from? Reference needed. A negative sign is missing at the CB energy level of g-C3N4 (−1.15 V).
There are several missing or redundant spaces in the manuscript.
The text of the manuscript omits a description of the conditions of several measurements (SEM, TEM, XPS, UV-vis spectroscopy, FT-IR, PL, applied lightsuorce).
The full manuscript must be revised and read by a native English speaker.
Author Response
"Please see the attachment."
